# Molecular and Functional Characterization of *Elovl4* Genes in *Sparus aurata* and *Solea senegalensis* Pointing to a Critical Role in Very Long-Chain (>C_24_) Fatty Acid Synthesis during Early Neural Development of Fish

**DOI:** 10.3390/ijms21103514

**Published:** 2020-05-15

**Authors:** Sofia Morais, Miguel Torres, Francisco Hontoria, Óscar Monroig, Inma Varó, María José Agulleiro, Juan Carlos Navarro

**Affiliations:** 1Instituto de Investigación y Tecnología Agroalimentaria (IRTA), Ctra. Poble Nou km 5.5, 43540 Sant Carles de la Rápita, Tarragona, Spain; sofia.morais@lucta.com (S.M.); mjagulleiro@gmail.com (M.J.A.); 2Instituto de Acuicultura de Torre de la Sal (IATS-CSIC), 12595 Ribera de Cabanes, Castellón, Spain; miguel.torres.rodriguez@csic.es (M.T.); oscar.monroig@csic.es (Ó.M.); inma@iats.csic.es (I.V.); jcnavarro@iats.csic.es (J.C.N.)

**Keywords:** Gilthead seabream, Senegalese sole, very long-chain polyunsaturated fatty acid, Elovl4, functional characterization, tissue expression, neural tissue development

## Abstract

Very long-chain fatty acids (VLC-FA) play critical roles in neural tissues during the early development of vertebrates. However, studies on VLC-FA in fish are scarce. The biosynthesis of VLC-FA is mediated by elongation of very long-chain fatty acid 4 (Elovl4) proteins and, consequently, the complement and activity of these enzymes determines the capacity that a given species has for satisfying its physiological demands, in particular for the correct development of neurophysiological functions. The present study aimed to characterize and localize the expression of *elovl4* genes from *Sparus aurata* and *Solea senegalensis*, as well as to determine the function of their encoded proteins. The results confirmed that both fish possess two distinct *elovl4* genes, named *elovl4a* and *elovl4b*. Functional assays demonstrated that both Elovl4 isoforms had the capability to elongate long-chain (C_20–24_), both saturated (SFA) and polyunsaturated (PUFA), fatty acid precursors to VLC-FA. In spite of their overlapping activity, Elovl4a was more active in VLC-SFA elongation, while Elovl4b had a preponderant elongation activity towards n-3 PUFA substrates, particularly in *S. aurata*, being additionally the only isoform that is capable of elongating docosahexaenoic acid (DHA). A preferential expression of *elovl4* genes was measured in neural tissues, being *elovl4a* and *elovl4b* mRNAs mostly found in brain and eyes, respectively.

## 1. Introduction

Certain long-chain (C_20–24_) polyunsaturated fatty acids (LC-PUFA), namely eicosapentaenoic acid (EPA; 20:5n-3), arachidonic acid (ARA; 20:4n-6) and docosahexaenoic acid (DHA; 22:6n-3), are regarded as physiologically essential for the correct development of vertebrates, including fish [1]. These compounds can be obtained through the diet or, alternatively, biosynthesized from C_18_ polyunsaturated fatty acids (PUFA), such as α-linolenic acid (18:3n-3) and linoleic acid (18:2n-6), via enzymatic reactions carried out by fatty acyl desaturases (Fads) and elongation of very long-chain fatty acid (Elovl) proteins [1,2]. Fads are enzymes that introduce double bonds (unsaturations) into PUFA substrates. On the other hand, Elovl are considered to be pivotal components of fatty acid (FA) synthetic pathways [3,4], being responsible for a condensation reaction, which results in the extension of the pre-existing FA chain with two new carbon atoms [1]. The Elovl protein family contains several members [2,3,4], of which only a few have been demonstrated to have PUFA as substrates. Of these, Elovl2, Elovl4, and Elovl5, have well-established roles in the biosynthesis of LC-PUFA in vertebrates [2,3,4], while a novel Elovl8 has been more recently suggested to be also involved in PUFA elongation [5,6]. While Elovl2 and Elovl5 are primarily involved in elongation steps within the LC-PUFA biosynthesis pathway, Elovl4 catalyzes the synthesis of very long-chain (>C_24_) PUFA (VLC-PUFA), which can have up to 36 or 38 carbons [1,7]. Furthermore, Elovl4 is additionally responsible for the production of very long-chain saturated fatty acids (VLC-SFA) [8].

Virtually all teleosts possess at least two Elovl4 isoforms termed as “Elovl4a” and “Elovl4b” [2,9]. Gene expression data indicates that both *elovl4* paralogs have widespread tissue distribution, with *elovl4a* being highly expressed in brain [9,10] and *elovl4b* in eye (retina) and gonads [9,10]. The functions of Elovl4a and Elovl4b seem to vary among species. For instance, in zebrafish (*Danio rerio*), Elovl4a showed the ability to elongate saturated FA (SFA) to produce VLC-SFA, while only Elovl4b was able to elongate PUFA substrates to produce VLC-PUFA [9]. However, studies performed on African catfish *(Clarias gariepinus*) [10] and black seabream (*Acanthopagrus schlegelii*) [11] have demonstrated that both Elovl4a and Elovl4b have the ability to biosynthesize VLC-PUFA. These results suggest that the investigation of Elovl4 proteins in teleosts requires a species-specific approach.

The gilthead seabream (*Sparus aurata*) and Senegalese sole (*Solea senegalensis*) are two commercially important species in marine finfish aquaculture. A recent study highlighed a relationship between the expression of *elovl4* genes in both species and the formation of neural tissues during early life-cycle development [12]. Indeed, Elovl4 products, i.e., VLC-SFA and VLC-PUFA, play crucial roles during the early-development of vertebrates by guaranteeing the correct development and functionality of the rapidly forming nervous system, where these compounds accumulate [8,12]. From what is known in higher vertebrates, VLC-PUFA are generally incorporated into phosphatidylcholine in the photoreceptor cells that make up the retina [7], and are then bioconverted into elovanoids, which participate in photoreceptor protection [8,13]. On the other hand, VLC-SFA are mainly incorporated into sphingolipids in the brain [10], taking part in the membrane fusion of synaptic vesicles that occur during the neurotransmission process in mammals [14,15]. Finally, Elovl4, including teleost Elovl4, can also play a role in the biosynthesis of LC-PUFA, specifically DHA [16,17], which is the most abundant FA in brain and retinal cells [18,19,20]. 

It is crucial to understand the capacity that a given species has for endogenous production of these essential nutrients due to the importance of very long-chain fatty acids (VLC-FA) during early development. Such ability is itself dependent on the complement of *elovl4* genes and the functions of their corresponding encoded enzymes [1]. Having this in mind, the aim of the present study was to characterize, both molecularly and functionally, *elovl4* paralogs from *S. aurata* (*Sa*) and *S. senegalensis* (*Ss*). Previous studies investigating the functions of *fads*- and other *elovl*-like genes confirmed that both species operate different LC-PUFA biosynthesis mechanisms [21,22,23,24], especially with regard to the production of DHA. In particular, *Sa* operates the so-called “Sprecher pathway” [21,25], whereas *Ss* produces DHA via the more direct “Δ4 pathway” [24] (Figure 1). We will discuss our results in the context of the biosynthetic particularities of both species when considering that both the LC-PUFA and VLC-PUFA biosynthetic pathways are interdependent.

## 2. Results

### 2.1. Elovl4 Sequence and Phylogenetic Analysis

The *Sa* and *Ss elovl4a* ORF sequences have 969 base pairs (bp) and 960 bp, encoding putative proteins of 322 amino acids (aa) and 319 aa, respectively (Appendix A). On the other hand, *Sa* and *Ss elovl4b* ORF sequences contain 918 bp, encoding proteins of 305 aa (Appendix A). Elovl4 from both *Sa* and *Ss* contain the conserved histidine binding box motif (HXXHH), the predicted endoplasmic reticulum (ER) retention signal with arginine (R) and lysine (K) at the carboxyl end (RXKXX), as well as several regions with conserved motifs (Appendix A). This suggests that *Sa* and *Ss elovl4* cDNA both encode Elovl4a and Elovl4b enzymes, which have been deposited in Genbank under the following accesion numbers: *Sa* Elovl4a (MK610320), *Sa* Elovl4b (MK610321), *Ss* Elovl4a (MN164537), and *Ss* Elovl4b (MN164625). We compared the deduced aa sequence of Elovl4a and Elovl4b from both species via BLASTp searches. The results revealed that *Sa* and *Ss* Elovl4a aa sequences are both 95% identical. Likewise, both *Sa* and *Ss* Elovl4b aa sequences show 90% identity with each other. The phylogenetic analysis showed that both Elovl4 sequences from each species form two separate clusters that include either Elovl4a or Elovl4b sequences from a range of teleosts (Figure 2). *Sa* and *Ss* Elovl4a both clustered together, while *Sa* and *Ss* Elovl4b clustered more closely with *A. schlegelii* Elovl4b and Atlantic bluefin tuna (*Thunnus thynnus*) Elovl4b, respectively (Figure 2). All fish Elovl4 elongases grouped with mammalian and birds orthologs, and separately from other vertebrate Elovl proteins, such as Elovl2 and Elovl5. 

### 2.2. Functional Characterization of Elovl4a and Elovl4b

The *Sa* and *Ss* putative Elovl4 elongases were functionally characterized in yeast *Saccharomyces cerevisiae* (strain InvSc1). Firstly, we investigated the ability of *Sa* and *Ss* Elovl4 to elongate VLC-SFA by comparing the SFA profiles of yeast that was transformed with the yeast expression vector pYES2 containing the *elovl4* coding regions, with those of control yeast transformed with empty pYES2, after incubation in the absence of exogenously added FA. The results confirmed that *Sa* and *Ss* Elovl4 isoforms are both involved in the biosynthesis of VLC-SFA (Table 1 and Table 2). More specifically, *Sa elovl4* transformed yeast showed a different (*p* ≤ 0.05) profile of SFA ≥ C_24_ when compared to control yeast, with decreased contents of 26:0 and increased levels of 28:0, 30:0, and 32:0 (Table 1). These results suggest that 26:0 is an important substrate for both *Sa* Elovl4 proteins, with Elovl4a appearing as the most active isoform, since the contents of all detected VLC-SFA longer than 26:0 were significantly higher as compared to the control (Table 1). On the other hand, *Ss elovl4* transformed yeast showed increased levels of 24:0 and 26:0, along with decreased levels of 28:0, 30:0, and 32:0 (Table 2). Therefore, the results suggest that the *Ss* Elovl4 proteins are involved in the biosynthesis of VLC-SFA up to 26:0, for which they use <C_24_ fatty acids as elongation substrates. 

Yeast transformed with both Elovl4 were incubated with C_18_ (18:4n-3 and 18:3n-6), C_20_ (20:5n-3 and 20:4n-6), and C_22_ (22:5n-3, 22:6n-3 and 22:4n-6) PUFA substrates in order to test the role that *Sa* and *Ss* Elovl4 elongases play in VLC-PUFA biosynthesis. It is commonly assumed that the fatty acid composition of yeast transformed with empty pYES2 (control) only consists of saturated and monounsaturated fatty acids, together with the corresponding exogenously added PUFA, since it is well established that *S. cerevisiae* possesses no PUFA elongation activity [26]. The chromatographic analyses of Elovl4-transformed yeast revealed that *Sa* Elovl4a elongated all n-6 PUFA substrates, i.e., 18:3n-6, 20:4n-6, and 22:4n-6 (Table 3), as well as n-3 PUFA substrates (18:4n-3, 20:5n-3, 22:5n-3, and 22:6n-3) (Table 3). *Sa* Elovl4b also elongated n-6 PUFA substrates, but showed particularly high affinity towards n-3 PUFA substrates, especially to elongate 22:6n-3 (DHA) substrates to 32:6n-3 (Table 3). Moreover, *Sa* Elovl4 elongases were both able to convert 20:5n-3 or 22:5n-3 to 24:5n-3, an intermediate substrate for DHA synthesis via the Sprecher pathway [25].

For Senegalese sole, both Elovl4 elongases presented the capability to elongate PUFA substrates from the n-3 (18:4n-3, 20:5n-3, 22:5n-3, and 22:6n-3) and n-6 (18:3n-6, 20:4n-6, and 22:4n-6) series, to longer chain FA of up to C_34_ (Table 4). As described above for *Sa*, *Ss* Elovl4b was able to convert 22:6n-3 (DHA) to VLC-PUFA of up to C_32_, an elongation capacity not exhibited by the *Ss* Elovl4a. It is noteworthy that *Ss* Elovl4b, and to a lesser extent Elovl4a, showed high capacity to elongate 20:5n-3 (EPA) to 22:5n-3 (Table 4), a key step that is required for DHA synthesis via the Δ4 pathway.

### 2.3. Tissue Expression of Elovl4 Genes

The tissue distribution of the two *elovl4* mRNA’s in gilthead seabream and Senegalese sole was analyzed by reverse transcription polymerase chain reaction (RT-PCR). In both species, *elovl4a* and *elovl4b* transcripts appear to be present in most of the analyzed tissues. In tissues of the gastrointestinal tract, liver, skin, or muscle, expression was only found with low signal intensity (Figure 3A,B). Although comparisons of transcript levels from RT-PCR analyses have to be made cautiously, for gilthead seabream, strong expression signals were found in the brain and gonad for *elovl4a*, and brain, eye, and gonad for *elovl4b* (Figure 3A). In the case of Senegalese sole, strong *elovl4a* expression was detected in the brain, with eye and brain having a high signal of *elovl4b* expression (Figure 3B).

Selected tissues of both fish species were then analyzed by real-time quantitative reverse transcriptase PCR (qPCR) in order to obtain quantitative values of *elovl4a* and *elovl4b* expression (Figure 3C–F). For gilthead seabream, significant differences (*p* ≤ 0.05) were found in *elovl4a* expression between all tissues, with the highest levels of mRNA in brain, followed by eye, and finally gonad (Figure 3C). For *elovl4b*, the highest expression was found in eye, while the brain and gonad showed significantly lower expression values (*p* ≤ 0.05) (Figure 3E). Similarly, a differential expression of *elovl4* genes was detected in the tissues of Senegalese sole (*p* ≤ 0.05). For *elovl4a*, significantly higher expression was found in brain, with a lower expression being measured in the eye and gonad (Figure 3D). In the case of *elovl4b*, the highest expression levels were detected in eye, with brain and gonad having significantly lower expression levels (Figure 3F).

## 3. Discussion

Sequence analyses revealed that the investigated predicted Elovl4 proteins contain all the characteristic domains of vertebrate Elovl4 family members [27,28], including the ER retention signal (RXKXX), the histidine box (HXXHH), which is involved in the electron transfer process during fatty acid elongation [4], and other transmembrane domains, similarly to what has been described in other fish species [9,10,11,29,30,31,32,33,34,35]. Furthermore, phylogenetic analysis confirmed the existence of two isoforms of *Sa* and *Ss* Elovl4, which clustered, together with corresponding Elovl4 orthologs from other teleosts, into different branches, thus confirming that the described Elovl4 isoforms are true orthologs of the Elovl4a and Elovl4b proteins that are present in teleosts [2]. The conservation of both Elovl4 isoforms in fish genomes [2,9,10,11,17,33] and their clear segregation into separate clusters points towards a likely functional specialization of these proteins in teleosts [30,31], which we aimed to further elucidate in this study by functionally characterizing the two isoforms in two new fish species with diverse life histories, dietary habits, and notably different LC-PUFA biosynthesis mechanisms [21,22,23,24].

Our results support the notion that both isoforms can participate in VLC-SFA elongation, as suggested in previous studies with other fish species including *D. rerio* [9], *C. gariepinus* [10] and Atlantic salmon (*Salmo salar*) [32]. Nevertheless, Elovl4a seems to be more efficient than Elovl4b at elongating VLC-SFA, similarly to what was reported in *D. rerio* [9]. This was particularly evident in *Sa*, where the comparison of the SFA profile of yeast transformed with *elovl4a* or *elovl4b* with that of control yeast revealed that Elovl4b was only active in the elongation step from 26:0 to 28:0, whereas Elovl4a was clearly able to significantly elongate SFA substrates from 26:0 up to 32:0. The functional characterization of *Ss* Elovl4 enzymes showed some differences with respect to *Sa*, particularly concerning the preferred fatty acid substrates. Whereas the *Sa* Elovl4 isoforms have 26:0 as the most preferred precursor for VLC-SFA biosynthesis, <C_24_ saturated fatty acids appear to be more adequate for the *Ss* Elovl4 isoforms. This could be related to differences in VLC-SFA requirements between the two fish species [12], but further studies are necessary in order to clearly establish this.

Functional analyses of Elovl4a and Elovl4b confirmed that both proteins actively participate in the biosynthesis of either n-6 or n-3 VLC-PUFA, from n-6 and n-3 PUFA substrates, in the two studied fish species. These results are in agreement with Elovl4 functional characterization studies in other fish species, which showed similar elongation capabilities [9,10,11,29,32,34,35], consistent with the functionality that is described in mammals [36] and other aquatic organisms, such as common octopus (*Octopus vulgaris*) [37]. However, intra- and inter-specific differences were found in the efficiency of the different Elovl4 isoforms to biosynthesize VLC-PUFA. In this respect, *Sa* Elovl4a showed a clearly higher affinity towards the elongation of n-6 PUFA substrates, while *Sa* Elovl4b was particularly active towards n-3 PUFA substrates. On the other hand, only Elovl4b was able to elongate DHA, up to 34:6n-3. Similar results were obtained in Senegalese sole, where both of the isoforms were active towards n-6 or n-3 PUFA substrates, in this case with less clear differences in terms of substrate preference, but only Elovl4b showed activity towards DHA.

Interestingly, although fish Elov4b seem to have a predominant role in the DHA biosynthesis pathway [10,11,29], both *Sa* Elovl4 isoforms had the ability to elongate 20:5n-3 and 22:5n-3 to 24:5n-3, which is a key intermediate FA in the biosynthesis of DHA via the ∆6 “Sprecher pathway” [25] occurring in this species [21,22]. Similarly, both *Ss* Elovl4 isoforms, although Elovl4b more prominently, had the capacity to produce 22:5n-3 from 20:5n-3. This is a key substrate for DHA biosynthesis via the ∆4-desaturation pathway that was carried out by *Ss* Fads2 [24]. This redundancy in an activity that is central for DHA biosynthesis [9,11,17,30,33,35] highlights the essentiality of this compound for the correct development and survival of marine fish [1]. It is well known that the correct biosynthesis of DHA is crucial for the normal development of the fish cognitive system, especially during early stages, when its deficiency can cause visual and/or neural damage [18,38]. Moreover, it is highly conceivable that the conservation of two Elovl4 enzymes with the ability to elongate 22:5n-3 and 24:5n-3 can confer a substantial adaptive advantage in marine fish species that have lost the *elovl2* gene during evolution [1,2,11].

It is also noteworthy that, similar to what has been described in zebrafish [9], *Sa* and *Ss* Elovl4a elongases both showed low elongation activity from DHA to 24:6n-3. This could suggest that, as described in rat retinas [39], EPA and not DHA might be the preferred substrate for VLC-PUFA biosynthesis in fish. This assumes the formation of VLC-PUFA hexaenes from LC-PUFA pentaenes via Δ6 desaturation of 24:5n-3, which should only take place in the case of gilthead seabream, since Senegalese sole is believed to lack this desaturation capacity in favour of a Δ4 Sprecher-independent pathway for DHA biosynthesis [24]. Paradoxically, our functional results revealed a higher capacity for 24:5n-3 production in this latter species. On the other hand, similarly to what has been found in other teleosts [10,11,29], Elovl4b proteins in both species were able to elongate 24:6n-3 up to 32:6n-3, a VLC-PUFA found in retinal phosphatidylcholine in fish [40,41]. Thus, this specific activity of Elovl4b proteins, along with the above-mentioned presence of 32:6n-3 in fish retina, is coherent with the tissue expression results obtained for both species, in which Elovl4b transcripts were mostly found in the eye suggesting that, similarly to what has been described in other teleosts, like *T. thynnus* [29], *D. rerio* [9], *A. schlegelli* [11], rainbow trout (*Oncorhynchus mykiss*) [31], *S. salar* [32], or orange-spotted grouper (*Epinephelus coioides*) [35]; this is a major tissue for VLC-PUFA biosynthesis.

The quantitative expression results confirmed previous evidences of a differential *elovl4a* and *elovl4b* tissue-specific expression pattern [12,42], with *elovl4a* being mostly expressed in fish brain [9,10,11,31], and *elovl4b* in eye [9,11,29,31,32,35]. Therefore, in spite of the specific functions of VLC-PUFA still not being fully understood in vertebrates [7], and their identification being very scarce in fish [40,41], the results reported here in terms of *elovl4* tissue expression in these two fish species suggests a role of these enzymes in the local biosynthesis and the incorporation of VLC-FA in fish neural tissues. This is in agreement with what is known in mammals [8], in which VLC-PUFA are key functional components, essential for the development and cell protection, of neural tissues such as those found in retina or brain [7,13,36]. More specifically, certain VLC-PUFA are synthesized and esterified at the *sn-1* position of the glycerol backbone of phosphatidylcoline, which is then deposited in retinal photoreceptors, where it plays an important neuroprotective role [7,8,43]. Other VLC-SFA are mainly incorporated into sphingolipids in the central nervous system [8], playing a key role in the membrane fusion of synaptic vesicles occurring during neurotransmission processes [14,15].

The application of this knowledge is of special relevance during early larval development, particularly in species with high commercial interest for aquaculture production, as is the case of gilthead seabream and Senegalese sole, and it should be kept in mind in feeding protocols during hatchery rearing. Mammalian *elovl4* expression is developmentally regulated in the brain, with expression peaking around the time of birth and falling as the brain matures [44], thus pointing to a prominent role of this protein in neurogenesis [8]. In fish species, it is equally likely that the optimal functioning of Elovl4 enzymes is particularly critical during early developmental stages, at a time when neural tissues are rapidly forming [20,45], in order to ensure correct biosynthesis and tissue accumulation of VLC-PUFA products [9,12,30,42]. Hence, not surprisingly, *elovl4* genes were found to be widely expressed in neural tissues (brain mass and eyes) during the embryonic phase of *D. rerio* [9] and cobia (*Rachycentron canadum*) [30]. Moreover, as previously described [12], retinogenesis in gilthead seabream and Senegalese sole larvae is clearly synchronized with an increase in expression of both *elovl4* genes. Consequently, as described in mammals [46], alterations in VLC-PUFA biosynthesis could negatively impact visual acuity and disrupt brain functionality, jeopardizing the normal development of fish. Although neural and visual structures of newly hatched fish larvae are undeveloped, cones and rod cells differentiate quite early [47,48], and their correct development and functionality is determining for fish larvae to begin feeding exogenously [49] and, hence, for their survivability. This is particularly relevant in visual predators, such as gilthead seabream and Senegalese sole, which previously showed a differential *elovl4* expression in larvae and postlarvae according to the VLC-PUFA putative needs associated with each life-stage and LC-PUFA dietary availability [12,42]. Finally, the results from the present study evidencing a low activity of Elovl4 on DHA and a higher activity on longer (26 and 28 C) substrates, which are, in turn, dependent on DHA, reinforce the idea that an appropriately high dietary supply of DHA is critical in early stages of fish larval life, not only “per se”, i.e., related to the essential nature of this fatty acid on its own, but also as a bottleneck substrate for subsequent VLC-PUFA synthesis.

In view of the results that are presented here, we conclude that both gilthead seabream and Senegalese sole possess two distinct Elovl4-like elongases named Elovl4a and Elovl4b based on their homology to the zebrafish orthologs [9]. Functional analyses denoted that, although with some specificities, both *Sa* and *Ss* Elovl4a and Elovl4b are involved in VLC-SFA and VLC-PUFA biosynthesis, being able to elongate a range of substrates up to C_34_ VLC-SFA and VLC-PUFA. Moreover, neural tissues are the major site of *elovl4* expression, with brain and eye exhibiting the highest *elovl4a* and *elovl4b* expression levels, respectively. Therefore, these are likely the main tissues of VLC-FA biosynthesis and accumulation, which highlights the importance of these compounds for crucial physiological processes, such as vision and brain function, particularly during early fish development.

## 4. Materials and Methods

All of the experimental procedures were conducted according to the European Union Directive (2010/63/EU) on the protection of animals for scientific purposes, at the Instituto de Acuicultura de Torre de la Sal (IATS-CSIC). The Animal Welfare and Bioethical Committee of IATS-CSIC approved all experimental conditions and sampling protocols under the code 015/2013 on 24 January 2014 according to Royal Decree RD53/2013.

### 4.1. Molecular Cloning of Elovl4 cDNA Sequences

Total RNA was isolated from gilthead seabream and Senegalese sole brain mass and eye using Maxwell 16 LEV simplyRNA Tissue Kit (Promega Biotech Ibérica S.L., Madrid, Spain). RNA quality and quantity were assessed by gel electrophoresis and spectrophotometry (NanoDrop ND-2000C, Thermo Fisher Scientific, Madrid, Spain). Two µg of total RNA from brain and eye were reverse transcribed into cDNA while using the M-MLV reverse transcriptase first strand cDNA synthesis kit (Promega Biotech Ibérica S.L.) following the manufacturer’s instructions, and while using a mixture (3:1, *v/v*) of random primers and anchored oligo (dT)_15_ primer (Promega Biotech Ibérica S.L.). Cloning of the *elovl4* full-length cDNA was carried out using PCR-based methodologies and brain-eye mix (1:1) cDNA as template. For *Sa elovl4* genes, degenerated primers UNIelovl4a-F/UNIelovl4a-R (*elovl4a*) and UNIelovl4b-F/UNIelovl4b-R (*elovl4b*), which were designed on conserved regions of teleost *elovl4a* and *elovl4b* orthologs available in the GenBank database, were used for amplification of the first fragment of the *Sa* putative *elovl4a* and *elovl4b* sequences with the PCR conditions that are shown in the Table 5. The PCR fragments were then purified while using the Illustra GFX™ PCR DNA and Gel Band Purification kit (GE Healthcare, Barcelona, Spain) and sequenced at least two times (DNA Sequencing Service, IBMCP-UPV, Valencia, Spain).

Two-round (nested) Rapid Amplification of cDNA ends (RACE) PCR were performed using the FirstChoice^®^ RLM-RACE kit (Ambion, Life Technologies, Madrid, Spain) on each 3′ and 5′ RACE cDNA synthesized from *Sa* brain and eye RNA following the manufacturer’s instructions in order to obtain the full-length ORF sequences. All of the primers used and PCR conditions are shown in Table 5. Potential positive fragments were cloned into pGEM-T Easy cloning vector (Promega Biotech Ibérica S.L.), while using GoTaq DNA polymerase (Promega Biotech Ibérica S.L.), and ligated with T4 DNA ligase (Promega Biotech Ibérica S.L.). The plasmid preparations were purified using the GenElute™ Plasmid Miniprep Kit (Sigma–Aldrich, Madrid, Spain), and sequenced as described above. Two putative *elovl4* sequences were thus obtained and deposited in the GenBank database as gb|MK610320 (*elovl4a*) and gb|MK610321 (*elovl4b*). All primers used in this assay were designed using Primer3 software (http://primer3.sourceforge.net) [50].

In the case of Senegalese sole, partial sequences were searched by gene annotation in the SoleaDB transcriptomic database (http://www.scbi.uma.es/soleadb) version 4.1 for *Ss* global assembly and then grouped and assembled in silico while using the BioEdit Sequence Alignment Editor (BioEdit v7.0.9; Tom Hall, North Carolina State University, Raleigh, NC, USA). A BLAST search was performed in the National Center for Biotechnology Information (NCBI) online database (http://www.ncbi.nlm.nih.gov/) to compare with orthologs in other fish and lower vertebrate species and identify the ORF and untranslated regions (UTR) of the sequences. In order to obtain the full-length ORF sequences, given that 3′ ends were missing but not 5′ ends, two-round (nested) RACE-PCR were performed using the FirstChoice^®^ RLM-RACE kit (Ambion, Life Technologies, Madrid, Spain) on 3′ RACE cDNA synthesized, as described above, from a 1:1 mix of Senegalese sole brain and eye RNA. The DNA fragments, which were amplified by PCR using GoTaq DNA polymerase (Promega Biotech Ibérica S.L.), were cloned into pGEM-T Easy (Promega Biotech Ibérica S.L.), purified using the GenElute™ Plasmid Miniprep Kit (Sigma–Aldrich), and then sequenced (DNA Sequencing Service, IBMCP-UPV), as above. Table 5 illustrates the primers used and PCR conditions. Two putative *elovl4* sequences were thus obtained and deposited in the GenBank database as gb|MN164537 (*elovl4a*) and gb|MN164625 (*elovl4b*).

### 4.2. Sequence and Phylogenetic Analysis

The BLAST sequence analysis of NCBI was used for sequence alignment and phylogenetic analysis. The aa sequences that were deduced from the nucleotide sequences of the *Sa* Elovl4a (gb|QES86604.1), *Ss* Elovl4a (gb|QGA31141.1), *Sa* Elovl4b (gb|QES86605.1), and *Ss* Elovl4b (gb|QGA31140.1), were aligned using the ClustalW tool (BioEdit v7.0.9). The percentage of identity among aa sequences were obtained by comparison of Elovl4a and Elovl4b from both species by using Standard Protein BLAST (NCBI). The phylogenetic tree was constructed while using a total of 31 aa sequences, including the herein characterized *Sa* and *Ss* Elovl4 enzymes and other fish and vertebrate Elovl4, Elovl2, and Elovl5 sequences, while using the Maximum Likelihood method and the JTT matrix-based model [51]. Confidence in the resulting phylogenetic tree branch topology was measured by bootstrapping through 1000 iterations. Phylogenetic analyses were conducted in MEGA X [52].

### 4.3. Functional Characterization of Sa and Ss Elovl4 Isoforms

The *Sa* and *Ss* putative Elovl4 elongases were functionally characterized by determining the FA profiles of *S. cerevisiae* (*S.c.* EasyComp™ Transformation Kit; Thermo Fisher Scientific) that were transformed with pYES2 yeast expression vector (Thermo Fisher Scientific) containing the putative *elovl4* as inserts, and grown in the presence of potential FA substrates. Briefly, PCR fragments corresponding to the ORF of the *Sa* and *Ss elovl4a* and *elovl4b* were amplified by nested PCR from a mixture of cDNA (brain and eyes) while using the high fidelity *Pfu* DNA polymerase (Promega Biotech Ibérica S.L.) with primers containing restriction sites, corresponding to *Eco*RI (forward) and *Xho*I (reverse) for the *Sa elovl4* genes, and *Hin*dIII (forward) and *Xho*I (reverse) for the *Ss elovl4* genes (Table 5). The obtained DNA fragments were purified, as described above, digested with the appropriate restriction enzymes (New England Biolabs Ltd., Hitchin, United Kingdom), and then ligated into a similarly restricted pYES2 yeast expression vector. The purified plasmids (GenElute™ Plasmid Miniprep kit, Sigma–Aldrich) containing the putative *elovl4* ORF sequences were used to transform *S. cerevisae* (strain InvSc1) competent cells. The transformation and selection of yeast with recombinant pYES2-*elovl4* or empty pYES2 (control) plasmids, and yeast culture were performed as described in detail previously [11].

Cultures of recombinant yeast were grown in *S. cerevisae* minimal medium minus uracil (SCMM^ura^, Sigma–Aldrich) broth supplemented with one of the following PUFA substrates: C_18_ (18:4n-3 and 18:3n-6), C_20_ (20:5n-3 and 20:4n-6), and C_22_ (22:5n-3, 22:6n-3 and 22:4n-6), in order to analyze the roles of *Sa* and *Ss* Elovl4 enzymes in VLC-PUFA biosynthesis. PUFA substrates were used at final concentrations 0.5 mM (C_18_), 0.75 mM (C_20_), and 1 mM (C_22_) to compensate for differential uptake related to FA acyl chain length [9]. Each PUFA substrate was tested once (*n* = 1). Recombinant yeast transformed with either pYES2 containing the corresponding *elovl4* ORF sequence or pYES2 with no insert (control) were grown in triplicate flasks (*n* = 3) in the absence of exogenously added FA to enable comparison of their long-chain (>C_24_) saturated FA profiles to study the ability of *Sa* and *Ss* Elovl4 enzymes to biosynthesize VLC-SFA. After two days at 30 °C, the yeast cells were harvested by centrifugation, washed twice in double distilled H_2_O, homogenized in 2:1 (*v/v*) chloroform:methanol containing 0.01% (*w/v*) butylated hydroxytoluene (BHT, Sigma–Aldrich) as antioxidant, and then stored at −20 °C until further analysis. Yeasts that were transformed with pYES2 containing no insert were cultured under the same conditions as a control treatment. All FA substrates (>98–99% pure) used for the functional characterization assays, except stearidonic acid (18:4n-3), were obtained from Nu-Chek Prep, Inc. (Elysian, MN, USA). Stearidonic acid (>99% pure) and *S. cerevisiae* culture reagents, including galactose, nitrogen base, raffinose, tergitol NP-40, and uracil dropout medium, were obtained from Sigma–Aldrich.

### 4.4. Fatty Acid Analysis

The total lipids were extracted from yeast samples while using the method that was described by Folch et al. [53]. Fatty acid methyl esters (FAME) were prepared, extracted, and purified, as described in detail in [12]. FAME were identified and quantified using a gas chromatograph (GC) coupled to a mass spectrometry (MS) detector, as described previously [12]. Briefly, the elongation of endogenous saturated FA with 24 carbons or longer was assessed by comparison of the areas of the fatty acids of control yeast with those of yeast that was transformed with each of the pYES2-*elovl4* plasmid constructs. The GC-MS was operated in the electron ionization (EI) single ion monitoring (SIM) mode. The 24:0, 26:0, 28:0, 30:0, 32:0, and 34:0 response values were obtained by using the *m*/*z* ratios 382.4, 410.4, 438.4, 466.5, 494.5, and 522.5, respectively. For VLC-PUFA analysis, the response values were obtained by using the *m*/*z* ratios 79.1, 108.1, and 150.1 in SIM mode [9,36,41]. In this case, the elongation conversions of exogenously added PUFA were calculated as (area of first product and longer chain products/(area of first product and longer chain products + substrate area)) × 100.

### 4.5. Tissue Expression of Elovl4 Genes in Gilthead Seabream and Senegalese Sole

#### 4.5.1. Sample Preparation

The samples for the tissue expression analysis of *elovl4* transcripts were obtained from three juveniles of gilthead seabream (14–17 cm; 40–60 g) and three Senegalese sole (16–19 cm; 60–80 g) that were maintained at the facilities of IATS-CSIC, and fed on standard diets. While samples of selected tissues from the three fish of each species were used for qPCR, all of the tissues from a single fish were used for the RT-PCR study. All fish were anesthetized with 3-aminobenzoic acid ethyl ester (MS-222, 100 μg/mL) and then quickly sacrificed by cervical dislocation before sample collection of tissues, including brain mass, eye, gonad, liver, stomach, intestine, skin, white muscle, and red muscle. All of the tissue samples were immediately frozen and stored at −80 °C until required for RNA isolation. The total RNA was isolated from the tissues and cDNA samples were prepared from 2 µg of total RNA, as described above.

#### 4.5.2. Gene Expression Analysis by Reverse Transcriptase PCR (RT-PCR)

The expression of *elovl4* isoforms in each tissue from one specimen of gilthead seabream and Senegalese sole was analyzed by reverse transcriptase PCR (RT-PCR) while using GoTag Polymerase (Promega Biotech Ibérica S.L.), using *18s ribosomal RNA* (*18s*) as a reference gene. PCR conditions consisted of an initial denaturing step at 95 °C for 2 min. followed by 35 cycles of denaturation at 95 °C for 30 s, annealing at 60 °C for 30 s, and extension at 72 °C for 35 s, ending with a final extension at 72 °C for 5 min. Table 6 shows the primers used for RT-PCR on tissue cDNA samples. The RT-PCR products were assessed by gel electrophoresis and photodocumented while using UV light in a Gel Documentation System Amersham Imager 600 (GE Healthcare UK Limited, Little Chalfont, UK). A random set of RT-PCR samples were purified and sequenced as above to confirm the identity of the amplicons.

#### 4.5.3. Gene Expression Analysis by Quantitative Real-Time PCR (qPCR)

The expression of *elovl4a* and *elovl4b* was analyzed by qPCR in tissues that showed a strong signal in RT-PCR analyses (brain, eye, and gonad), from three fish. Table 6 shows the primers used in qPCR analyses. The efficiency of the primer pairs was assessed through a standard curve that was obtained by serial dilutions of standard solutions of the studied genes with known copy numbers, which also allowed for the conversion of threshold cycle (Ct) values to copy numbers. The amplification was carried out, as previously described in [12]. Three potential reference genes (*β*-*actin, elongation factor 1α* and *18s rRNA*) were tested. After checking gene stability using the Genorm software [54], *β-actin* was chosen for gene expression normalization. The gene expression results are given as mean normalized values ± standard deviation (SD) corresponding to the ratio between copy numbers of fatty acyl elongase (*elovl4a* and *elovl4b*) transcripts and copy numbers of the reference gene *β-actin* (*actb*).

### 4.6. Statistical Analysis

The homogeneity of variances of the data that were associated to VLC-SFA (%) and tissue gene expression values, as determined by qPCR, were checked using Levene’s test. Statistical differences were analyzed by one-way analysis of variance (ANOVA) (*p* ≤ 0.05), followed by Tukey HSD post-hoc tests. The statistical software SPSS 26.0 (SPSS Inc., Chicago, IL, USA) was used to analyze the data.

## Figures and Tables

**Figure 1 ijms-21-03514-f001:**
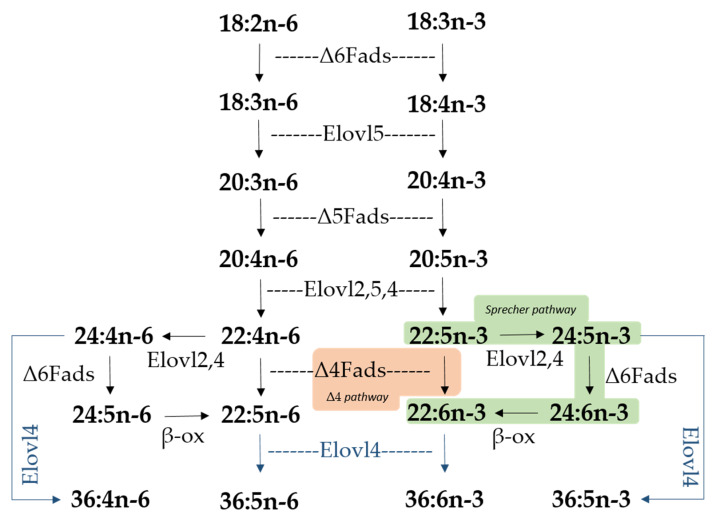
Biosynthetic pathways of long-chain (LC-PUFA; C_20–24_) and very long-chain polyunsaturated fatty acids (VLC-PUFA; >C_24_) in fish. Desaturation reactions are mediated by fatty acyl desaturases (Fads), whereas elongation reactions are catalyzed by elongation of very long-chain fatty acid (Elovl) proteins. Microsomal β-oxidation reactions are denoted as “β-ox”. Two pathways for docosahexaenoic acid (DHA; 22:6n-3) biosynthesis from docosapentaenoic acid (DPA; 22:5n-3) are indicated, namely the Sprecher pathway (green background) and the Δ4 pathway (orange background). Elongation reactions leading to VLC-PUFA biosynthesis of up to C_36_ are indicated with blue arrows. Note the fish species studied herein (*Sparus aurata* and *Solea senegalensis*) lack *elovl2* in their genomes [2].

**Figure 2 ijms-21-03514-f002:**
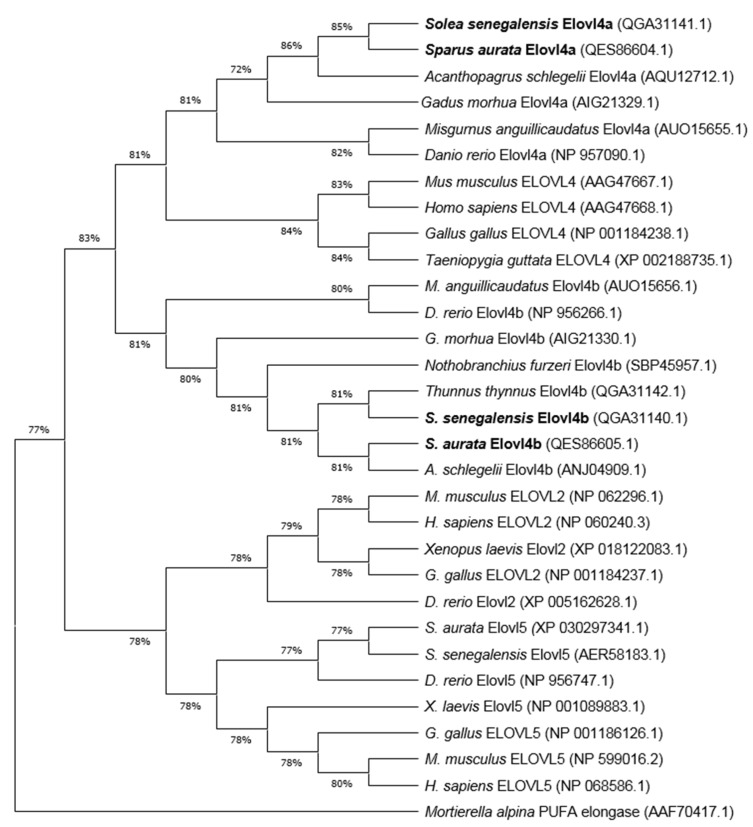
Phylogenetic tree comparing *Sparus aurata* and *Solea senegalensis* Elovl4a and Elovl4b proteins (highlighted in bold) with Elovl2, Elovl4 and Elovl5 proteins from other vertebrates. The tree was constructed while using the Maximum Likelihood method and Jones-Taylor-Thornton (JTT) matrix-based model. The numbers in branches represent the frequencies (%) of each node after 1000 iterations by bootstrapping. The *Mortierella alpina* PUFA elongase was included in the analysis as an outgroup, to construct the rooted tree.

**Figure 3 ijms-21-03514-f003:**
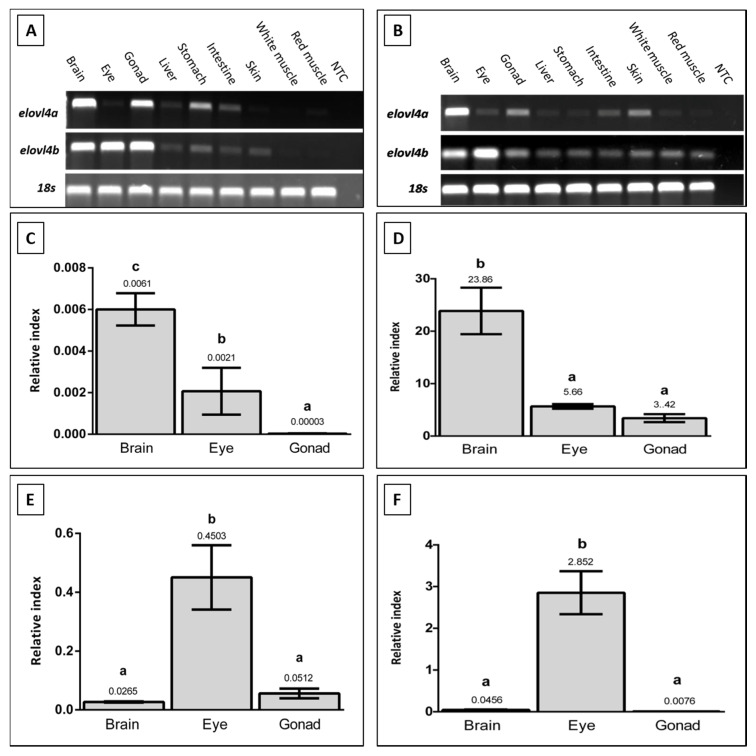
Tissue distribution of *elovl4a* and *elovl4b* transcripts in *Sparus aurata* (**A**) and *Solea senegalensis* (**B**) determined by Reverse Transcriptase-Polymerase Chain Reaction (RT-PCR) (*n* = 1 fish). Expression of housekeeping gene 18s is also shown. Expression in selected tissues of *Sparus aurata elovl4a* (**C**) *Solea senegalensis elovl4a* (**D**), *Sparus aurata elovl4b* (**E**) and *Solea senegalensis elovl4b* (**F**) transcripts were also determined by qPCR. The results, shown as relative index, are β-actin normalized values (gene copy number/*β-actin* copy number). Bars represent means and standard deviations (*n* = 3 fish). Different letters denote significant differences (ANOVA and Tukey HSD test, *p* ≤ 0.05) among tissues.

**Table 1 ijms-21-03514-t001:** Functional characterization of *Sparus aurata* Elovl4 elongases: role in the biosynthesis of very long-chain saturated fatty acids (FA).

FA	Elovl4a	Elovl4b	Control
24:0	14.6 ^a^	11.3 ^a^	12.1 ^a^
26:0	49.5 ^b^	68.2 ^a^	75.0 ^a^
28:0	20.8 ^c^	14.1 ^b^	8.6 ^a^
30:0	11.0 ^b^	4.5 ^a^	2.7 ^a^
32:0	3.3 ^b^	1.5 ^a^	1.0 ^a^
34:0	0.7 ^a^	0.3 ^a^	0.3 ^a^

The results are expressed as area percentage (%) of total saturated FA ≥ C_24_ found in yeast transformed with either pYES2 containing the *elovl4* coding regions or empty pYES2 vector (control) (*n* = 3). Different superscripts denote significant differences in each row, among area percentages of each saturated FA (one way-ANOVA and Tukey test, *p* ≤ 0.05).

**Table 2 ijms-21-03514-t002:** Functional characterization of *Solea senegalensis* Elovl4 elongases: role in the biosynthesis of very long-chain saturated fatty acids (FA).

FA	Elovl4a	Elovl4b	Control
24:0	9.2 ^b^	9.5 ^b^	6.1 ^a^
26:0	72.1 ^b^	81.2 ^b^	58.3 ^a^
28:0	11.9 ^b^	5.7 ^c^	21.7 ^a^
30:0	5.5 ^b^	2.9 ^b^	11.5 ^a^
32:0	1.4 ^ab^	0.7 ^b^	2.4 ^a^
34:0	0.0 ^a^	0.0 ^a^	0.0 ^a^

Results are expressed as area percentage (%) of total saturated FA ≥ C_24_ found in yeast transformed with either pYES2 containing the *elovl4* coding regions or empty pYES2 vector (control) (*n* = 3). Different superscripts denote significant differences in each row, among area percentages of each saturated FA (one way-ANOVA and Tukey test, *p* ≤ 0.05).

**Table 3 ijms-21-03514-t003:** Functional characterization of the *Sparus aurata* Elovl4a and Elovl4b elongases by heterologous expression in the yeast *Saccharomyces cerevisiae*.

FA Substrate	Product	Elovl4a	Elovl4b
		% Conversion	% Conversion
18:4n-3	20:4n-3	2.5	2.7
	22:4n-3	9.7	12.5
	24:4n-3	5.6	49.9
	26:4n-3	n.d.	65.6
	28:4n-3	n.d.	n.d.
	30:4n-3	n.d.	n.d.
	32:4n-3	n.d.	n.d.
	34:4n-3	n.d.	n.d.
	36:4n-3	n.d.	n.d.
18:3n-6	20:3n-6	2.6	2.1
	22:3n-6	21.6	9.6
	24:3n-6	52.5	n.d.
	26:3n-6	57.1	n.d.
	28:3n-6	64.8	n.d.
	30:3n-6	90.0	n.d.
	32:3n-6	84.1	n.d.
	34:3n-6	41.3	n.d.
	36:3n-6	n.d.	n.d.
20:5n-3	22:5n-3	5.8	9.1
	24:5n-3	17.2	33.3
	26:5n-3	20.0	57.8
	28:5n-3	n.d.	86.8
	30:5n-3	n.d.	97.7
	32:5n-3	n.d.	72.7
	34:5n-3	n.d.	8.1
	36:5n-3	n.d.	n.d.
20:4n-6	22:4n-6	10.9	8.9
	24:4n-6	31.0	30.2
	26:4n-6	37.1	55.9
	28:4n-6	39.0	81.0
	30:4n-6	88.6	37.8
	32:4n-6	83.6	n.d.
	34:4n-6	73.7	n.d.
	36:4n-6	11.4	n.d.
22:5n-3	24:5n-3	3.4	12.6
	26:5n-3	19.8	52.2
	28:5n-3	26.0	86.3
	30:5n-3	85.6	96.5
	32:5n-3	74.2	64.4
	34:5n-3	63.0	5.3
	36:5n-3	n.d.	n.d.
22:4n-6	24:4n-6	8.2	10.4
	26:4n-6	35.1	43.1
	28:4n-6	45.5	71.8
	30:4n-6	90.8	83.0
	32:4n-6	78.7	19.5
	34:4n-6	54.6	n.d.
	36:4n-6	7.2	n.d.
22:6n-3	24:6n-3	0.4	1.8
	26:6n-3	n.d.	100
	28:6n-3	n.d.	100
	30:6n-3	n.d.	40.2
	32:6n-3	n.d.	61.3
	34:6n-3	n.d.	n.d.
	36:6n-3	n.d.	n.d.

The data are presented as the percentage conversions of polyunsaturated fatty acid (FA) substrates (*n* = 1)**.** Individual conversions were calculated according to the formula (area of first product and longer chain products/(area of first product and longer chain products + substrate area)) × 100. n.d.: not detected.

**Table 4 ijms-21-03514-t004:** Functional characterization of the *Solea senegalensis* Elovl4a and Elovl4b elongases by heterologous expression in the yeast *Saccharomyces cerevisiae*.

FA substrate	Product	Elovl4a	Elovl4b
		% Conversion	% Conversion
18:4n-3	20:4n-3	4.5	8.1
	22:4n-3	19.6	41.2
	24:4n-3	39.5	79.0
	26:4n-3	39.6	95.3
	28:4n-3	100	96.8
	30:4n-3	100	98.7
	32:4n-3	65.4	65.7
	34:4n-3	n.d.	1.7
	36:4n-3	n.d.	n.d.
18:3n-6	20:3n-6	4.6	6.2
	22:3n-6	38.6	40.8
	24:3n-6	66.2	66.0
	26:3n-6	65.3	89.1
	28:3n-6	100	91.9
	30:3n-6	55.0	90.4
	32:3n-6	62.7	17.8
	34:3n-6	n.d.	n.d.
	36:3n-6	n.d.	n.d.
20:5n-3	22:5n-3	12.1	30.9
	24:5n-3	31.8	75.1
	26:5n-3	35.7	87.4
	28:5n-3	100	96.9
	30:5n-3	50.0	98.9
	32:5n-3	33.7	82.9
	34:5n-3	38.2	14.5
	36:5n-3	n.d.	n.d.
20:4n-6	22:4n-6	18.1	33.1
	24:4n-6	49.9	73.4
	26:4n-6	56.7	85.1
	28:4n-6	65.2	94.3
	30:4n-6	95.2	95.9
	32:4n-6	84.9	51.8
	34:4n-6	25.3	2.7
	36:4n-6	n.d.	n.d.
22:5n-3	24:5n-3	7.8	44.3
	26:5n-3	33.9	87.9
	28:5n-3	51.2	97.0
	30:5n-3	92.3	99.0
	32:5n-3	27.4	82.5
	34:5n-3	32.4	16.2
	36:5n-3	n.d.	n.d.
22:4n-6	24:4n-6	13.5	37.2
	26:4n-6	58.3	85.5
	28:4n-6	71.8	94.5
	30:4n-6	94.5	96.3
	32:4n-6	21.6	53.9
	34:4n-6	25.9	5.0
	36:4n-6	n.d.	n.d.
22:6n-3	24:6n-3	0.6	5.1
	26:6n-3	n.d.	100
	28:6n-3	n.d.	100
	30:6n-3	n.d.	100
	32:6n-3	n.d.	22.3
	34:6n-3	n.d.	n.d.
	36:6n-3	n.d.	n.d.

Data are presented as the percentage conversions of polyunsaturated fatty acid (FA) substrates (*n* = 1). Individual conversions were calculated according to the formula (area of first product and longer chain products/(area of first product and longer chain products + substrate area)) × 100. n.d.: not detected.

**Table 5 ijms-21-03514-t005:** Nucleotide sequences of primers (Forward: F; Reverse: R) used for DNA open reading frame (ORF) cloning of *Sparus aurata* and *Solea senegalensis elovl4a* and *elovl4b*.

***Sparus aurata***					
**Aim**	**Primer**	**Sequence (5′–3′)**	**Ta**	**PCR Cycles**	**Extension Time**
**First fragment**	UNIelovl4a-F	TGATGGACAACCCCCTGC	57 °C	35	1 min
	UNIelovl4a-R	GCAGATGAGGGAGTAGTGCAT	57 °C	35	1 min
	UNIelovl4b-F	ATGGAGCCTTACTATAGCAGAC	55 °C	35	1 min
	UNIelovl4b-R	GCGAAGAGGATGATGAAGGT	55 °C	35	1 min
**5′ RACE PCR**	SaE4a-5R-R1	TTCTTCATGTACTTGGGCCC	60 °C	32	2 min 30 s
	SaE4a-5R-R2	AGAGGAACAGCAGGTAGGAGG	60 °C	32	2 min 30 s
	SaE4b-5R-R1	AGGTACAGGCAGCTGATGG	58 °C	32	2 min 30 s
	SaE4b-5R-R2	GAGATGACATCATGGGCCA	60 °C	32	2 min 30 s
**3′ RACE PCR**	SaE4a-3R-F1	GTGGACCCAAGATCCAGAAG	60 °C	32	2 min 30 s
	SaE4a-3R-F2	TGTCCCTCTACGTCAACTGC	60 °C	32	2 min 30 s
	SaE4b-3R-F1	TACCTCACCATCATCCAGATG	58 °C	32	2 min 30 s
	SaE4b-3R-F2	CTCTACACAGGCTGCCCATT	60 °C	32	2 min 30 s
**ORF Cloning**	SaE4a-U-F1	GATCTTTAAAGCGCCGACAC	56 °C	32	2 min 40 s
	SaE4a-U-R1	TCCGGCTAAATCTTCCTCAA	56 °C	32	2 min 40 s
	SaE4a-V-F2	CCCGAATTCACCATGGAGATTGTCACACA	60 °C	32	2 min
	SaE4a-V-R2	CCGCTCGAGCTCTAATCTCTTTTAGCCCTT	60 °C	32	2 min
	SaE4b-U-F1	AATCGAGACCAAAGGCAGAG	56 °C	32	2 min 40 s
	SaE4b-U-R1	CTCTGTTAATCGCCGAGCAC	56 °C	32	2 min 40 s
	SaE4b-V-F2	CCCGAATTCACCATGGAGGTTGTAACACA	60 °C	32	2 min
	SaE4b-V-R2	CCGCTCGAGCCTCTTCCTTCTTTACTCCC	60 °C	32	2 min
***Solea senegalensis***					
**Aim**	**Primer**	**Sequence (5′–3′)**	**Ta**	**PCR Cycles**	**Extension Time**
**3′ RACE PCR**	SsE4a-3R-F1	GGAGGAGAAAGAGGAAAGG	60 °C	35	2 min 30 s
	SsE4a-3R-F2	GAAAGGAAGAGCTAAAAGAGA	60 °C	35	2 min 30 s
	SsE4b-3R-F1	CGGTCACCTTCATCATCCTC	60 °C	35	2 min 30 s
	SsE4b-3R-F2	ATGCCTTCCTACACCCAGAA	60 °C	35	2 min 30 s
**ORF Cloning**	SsE4a-U-F1	ACTGGATCACGACCACAACC	55 °C	32	2 min 15 s
	SsE4a-U-R1	TCCCAACACAGGCACATCTC	55 °C	32	2 min 15 s
	SsE4a-V-F2	CCCAAGCTTACCATGGAGATTGTCACACATTTA	55 °C	32	2 min
	SsE4a-V-R2	CCGCTCGAGTTAATCTCTTTTAGCTCTTCCTTTC	55 °C	32	2 min
	SsE4b-U-F1	CGGGGAGGAGGAGAAGAAGA	55 °C	32	2 min 15 s
	SsE4b-U-R1	AGCAATCCCCTTGACCGTTT	55 °C	32	2 min 15 s
	SsE4b-V-F2	CCCAAGCTTACCATGGAGGTTGTAACACATTTTG	55 °C	32	2 min
	SsE4b-V-R2	CCGCTCGAGTTACTCTCTTTTGGCTCTTCCTT	55 °C	32	2 min

PCR parameters, annealing temperatures (Ta), number of cycles (PCR cycles) and extension time, are shown. Restriction sites (*Eco*RI and *Xho*I for *S. aurata*; *Hin*dIII and *Xho*I for *S. senegalensis)* in primers used for cloning into yeast expressions vector pYES2 are underlined.

**Table 6 ijms-21-03514-t006:** Primers used for reverse transcriptase PCR (RT-PCR) and real-time quantitative PCR (qPCR) of *Sparus aurata* and *Solea senegalensis* genes.

***Sparus aurata***						
**Aim**	**Transcript**	**Primer**	**Primer Sequence (5′–3′)**	**Ta**	**Fragment**	**Accession No**
**RT-PCR**	*elovl4a*	F	GCCCAAGTACATGAAGAACAGAG	60 °C	563 bp	MK610320
		R	GGGAGTAGTGCATCCAGTGG			
	*elovl4b*	F	GTCAAGTACTCCAACGATGTCAA	60 °C	394 bp	MK610321
		R	GGAATGGGCAGCCTGTGT			
	*18s*	F	TCCTTTGATCGCTCTACCGT	60 °C	460 bp	AY993930.1
		R	TGCCCTCCAATTGATCCTCG			
**qPCR**	*elovl4a*	F	GCCCAAGTACATGAAGAACAGAG	60 °C	169 bp	MK610320
		R	ACCTGATGAGTCTGCTGGGG			
	*elovl4b*	F	GTCAAGTACTCCAACGATGTCAA	60 °C	247 bp	MK610321
		R	GAGAAGGTAGGTACACGAGT			
	*Actb*	F	TGCGTGACATCAAGGAGAAG	60 °C	190 bp	X89920
		R	AAGGAGCCATACCTCAGGAC			
***Solea senegalensis***						
**Aim**	**Transcript**	**Primer**	**Primer Sequence (5′–3′)**	**Ta**	**Fragment**	**Accession No**
**RT-PCR**	*elovl4a*	F	TGCACTACTCCCTCATCTGC	60 °C	497 bp	MN164537
		R	TGAAAACAGCCACCTTAGGC			
	*elovl4b*	F	CCTCTGCCTTGTCCAGTTTC	60 °C	175 bp	MN164625
		R	TCCTTGACCCGTAGTTTAAC			
	*18s*	F	TCAGACCCAAAACCCATGCG	60 °C	464 bp	EF126042.1
		R	CCCGAGATCCAACTACGAGC			
**qPCR**	*elovl4a*	F	AGGTGAGGTAGGGCCTTGTT	60 °C	220 bp	MN164537
		R	CGGATTCCACCGACAAAAGT			
	*elovl4b*	F	CCTCTGCCTTGTCCAGTTTC	60 °C	175 bp	MN164625
		R	TCCTTGACCCGTAGTTTAAC			
	*Actb*	F	ACAATGAGCTGAGAGTCGCC	60 °C	132 bp	DQ485686
		R	ATGGGGGCGGTACATACAAC			

Sequences of primer pairs used (Forward: F; Reverse: R), annealing temperatures (Ta) of primer pairs, size of fragments produced, and accession number of the sequences used for primer design are shown.

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
