# Peer review of "Molecular and Functional Characterization of Elovl4 Genes in Sparus aurata and Solea senegalensis Pointing to a Critical Role in Very Long-Chain (>C24) Fatty Acid Synthesis during Early Neural Development of Fish"

_ijms, 2020, doi:10.3390/ijms21103514_

Round 1

Reviewer 1 Report

In this manuscript entitled “Molecular and functional characterization of elovl4 genes in Sparus aurata and Solea senegalensis pointing to a critical role in very long-chain (>C24) fatty acid synthesis during early neural development of fish” (Ref: ijms-794564) by Morais and colleagues, the authors aimed to characterize and localize elovl4 genes in both fishes and investigated the function of the encoded proteins in the biosynthesis of VLC-FAs. Overall, the data are well presented. Nevertheless, there are some criticism regarding some parts of the manuscript. A revision of these points therefore seems necessary to me. 

The following points should be considered:

General:

  • When naming a species, it would be nice to include the trivial name in addition to the Latin name when the species is first mentioned in the text.
  • RT-PCR and qPCR analysis: When I compare the results of both analyses, it is noticeable that there is a different ranking of the tissues with regard to the elov4l gene expression. For example: while by RT-PCR analysis elov4l in Sa is top expressed in 1) brain, 2) gonad and 3) stomach and furthermore the expression in the eye is very low, the order determined by qPCR is 1.) brain, 2.) eye, 3.) gonad. How can that be? In addition I would suggest to display the relative copy numbers if you want to add quantitative values since otherwise I don´t see the added value of this analysis compared to the RT-PCR.

Abstract:

  • Line 29: No explanation for the abbreviation DHA

Results:

  • Lines 98 to 100: Please check the length of the given coding sequences and the respective amino acid sequences since they seem to be wrong. They do not match with the database entries.
  • Figure 2: Please improve the quality since the figure is blurry. Furthermore, the figure should be moved to the supplementary section since its additional significance is rather limited.
  • Figure 2 legend: The text should not be in italic letters except the species names.
  • Line 160: Please include the data in the supplementary section or remove respective information from the text.
  • Line 191 and Figure 4: For Sa elovl4a there seems to be no signal in the white muscle.

Discussion:

  • Line 223: please use the term “predicted proteins”
  • Line 229: please use “isoforms” instead of “Elov14`s”
  • Line 229 and 329: They are not only true orthologs of the zebrafish proteins. Why do you hust pick this species?
  • Line 235 to 273: please revise this section since some parts are redundant and whole part can be shortened.
  • Line 276: “EPA and not DHA is the preferred substrate for VLC-PUFA biosynthesis in fish.” I think it is too speculative to draw the same conclusions for all fishes from the results in 3 fish species. Please revise.

Material and Methods:

  • It is unclear how many fish were used for the RT-PCR and qPCR analysis.
  • Line 356: Please indicate how many times you sequenced each nucleotide position at least
  • Lines 358 to 383: Redundant in large areas. Perhaps combine?
  • Line 383: include the Abbreviation for “GenBank”
  • Line 397: which 31 aa were used for the analysis? A domain or just randomly picked?
  • 3: Did you use replicates, triplicates…?
  • 5.2: What was the amount of RNA you reverse transcribed of each tissue?

Reviewer 2 Report

In their submitted manuscript Morais and coworkers present an analysis of elongation of very long-chain fatty acid 4 (Elovl4) orthologs found in two commercially important fish species. As Elovl4 and similar proteins that play an important role in fatty-acid homeostasis are important in the neural development of vertebrates their impairment can have profound consequences on the wellbeing of fish.

The submitted manuscript makes an important contribution, and clearly fits the Special Issue where it is intended, but I have a few suggestions regarding the data and its presentation.

  1. On Figure 2 the alignment should include Elovl4 proteins from other species, where the conservation of these sequences can be seen even better. I would suggest including zebrafish, mice and human, the very least. Also at this point black colour is clearly marking some non-identical amino acids as well, despite the figure legend claiming otherwise.
  2. In Tables 1 and 2 the superscripts indicating statistical significance should be explained in the table legends. It is also not clear that data from how many samples is presented here (n=?) and I am not convinced that the table format is the best way to present this data. Even if n is low, the data should be presented in the form of data points in a chart, so readers can see the variation. I bring this up, as clearly the control strains in Table 1 and 2 show very different results, suggesting variability.
  3. It is also not clear if the data for Table 3 and 4 are from single experiments or not. I am also a bit confused by the stated formula ([area of first product and longer chain products / (areas of all products with longer chain than substrate + substrate area)] x 100) as this, in my interpretation, would mean that we should see decreasing values in each column, which is clearly not the case. Could you please elaborate on this?
  4. Figures 4-7 could be compiled in a single, multi-panel figure. For the data presented in Figures 6 and 7 please do not use bar charts, as it is not an adequate form of representing this type of data (see in detail here: https://thenode.biologists.com/leaving-bar-five-steps/research/). For n=3 I would suggest displaying individual data points.
  5. The results on Figures 6 and 7 are somewhat surprising as they seem to contradict the semi-quantitative data on Figures 4 and 5. E.g. on Figure 4 S. aurata elovl4a expression seems to be much higher in the gonad than in the eye, whereas Figure 6 qPCRs clearly suggests a different picture. The same seems to be true for S. senegalensis elolvl4a (Figures 5 vs 7). Could you please comment on this?
